# The Potential of Universal Primers for Barcoding of Subtropical Crops: Actinidia, Feijoa, Citrus, and Tea

**DOI:** 10.3390/ijms26146921

**Published:** 2025-07-18

**Authors:** Lidiia S. Samarina, Natalia G. Koninskaya, Ruset M. Shkhalakhova, Taisiya A. Simonyan, Gregory A. Tsaturyan, Ekaterina S. Shurkina, Raisa V. Kulyan, Zuhra M. Omarova, Tsiala V. Tutberidze, Alexey V. Ryndin, Yuriy L. Orlov

**Affiliations:** 1Federal Research Centre the Subtropical Scientific Centre of the Russian Academy of Sciences, 354002 Sochi, Russia; natakoninskaya@mail.ru (N.G.K.); shhalahova1995@mail.ru (R.M.S.); taisiya-simony@yandex.ru (T.A.S.); grisha.tsaturyan@yandex.com (G.A.T.); shurkinaekaterina@mail.ru (E.S.S.); raisa.kulyan22@gmail.com (R.V.K.); zuly_om@mail.ru (Z.M.O.); tutberidse_tsiala@mail.ru (T.V.T.); subplod@mail.ru (A.V.R.); 2Agrarian and Technological Institute, Peoples’ Friendship University of Russia, 117198 Moscow, Russia

**Keywords:** Acca, kiwi, tea, citrus, barcode, intraspecific diversity, fingerprinting, chloroplast DNA, internal transcribed spacer

## Abstract

The molecular identification of valuable genotypes is an important problem of germplasm management. In this study, we evaluated the potential of 11 universal primer pairs for the DNA barcoding of locally derived cultivars of subtropical crops (actinidia, feijoa, citrus, and tea). A total of 47 accessions (elite cultivars, forms, and breeding lines) of these four genera were included in the study. The efficiency of the following universal primers was assessed using Sanger sequencing: ITS-p5/ITS-u4, ITS-p5/ITS-u2, ITS-p3/ITS-u4, 23S,4.5S&5S, 16S, petB/petD, rpl23/rpl2.l, rpl2 intron, rpoC1 intron, trnK intron, and trnE-UUC/trnT-GUU. Among these primers, trnE-UUC/trnT-GUU showed greater intraspecific polymorphisms, while rpl2 intron and 16S displayed the lowest polymorphism levels in all crops. In addition, the 23S,4.5S & 5S, and rpoC1 intron were efficient for intraspecific analysis of tea and actinidia species. Using five efficient chloroplast primers, a total of 22/6 SNPs/InDels were observed in tea accessions, 45/17 SNPs/InDels in actinidia, 23/3 SNPs/InDels in mandarins, and 5/4 SNPs/InDels in feijoa. These results will be useful for the further development of DNA barcodes of related accessions.

## 1. Introduction

Germplasm collections outside of the main producing regions are an important source of germplasm for global crop improvement. Border growing regions can be useful for providing new genetic resources with a wider genetic base and a higher adaptability to changing environmental conditions. Local genotypes and landraces in border growing regions offer a broad range of resilience to different agro-climatic conditions and are, therefore, an important resource for breeding purposes. Deep germplasm characterization and analysis of genetic diversity in these regions can shed light on the molecular mechanisms of plant domestication in extreme environments and are necessary for the appropriate utilization of genetic resources [1,2,3].

The northwestern coast of Caucasus is the northernmost boarder growing region for many subtropical crop species. Collections of subtropical crops, including feijoa, citrus, tea, and actinidia, were developed there based on plant material introduced from Japan, China, Sri Lanka, India, Italy, and other countries in the early 1900s [4,5]. Breeding efforts have been performed since the early 1930s; conventional breeding (mostly controlled hybridization, clonal selection) has resulted in the development of sets of cultivars and breeding lines that are characterized by exceptional fruit and leaf quality and abiotic stress tolerance [6,7]. Many cultivars and genotypes of subtropical crops are difficult to distinguish by their morphological traits. Therefore, the development of an efficient molecular method of identification for valuable genotypes will be useful for their proper characterization and utilization.

DNA barcoding, the method of characterizing species using one or a few standardized regions of DNA [8], has been used to both characterize existing biodiversity and identify new or cryptic species [9]. The development of DNA barcodes for local germplasm accessions is important to trace the movement of germplasm for the conservation of plants, especially since biodiversity has been threatened by anthropogenic activity [10,11]. Several approaches to plant DNA barcoding have been recently established using a target region of nuclear DNA (ITS-region), mitochondrial DNA, and chloroplast DNA (rbcL and matK genes, trnH-psbA, rpoB, and rpoC1 et al.) [12,13,14,15]. Among them, the nuclear region usually exhibits a higher level of polymorphism, while chloroplast regions show lower levels of polymorphism. However, for highly heterozygous crops, such as tea crops, the amplification and analysis of chloroplast fragments can be an easier approach as compared to nuclear DNA-fragments. In earlier studies, DNA barcoding was based on the size diifferences of the amplified fragments. Meanwhile, an analysis of the genetic variations inside of the fragments provides a greater opportunity to reveal intraspecific genetic polymorphisms in crops. Therefore, the goal of this study was to assess the potential of a set of universal primers for molecular identification of valuable local genotypes of subtropical crops (tea, citrus, actinidia, and feijoa). The results may be useful for the further management of the collections of these crops.

## 2. Results

Among the eleven primer pairs, two (trnK intron, petB/petD) did not result in the amplification of fragments. In addition, primers targeting the ITS region generated high-quality sequences and polymorphic sites in actinidia only. In tea plants, these primers generated no bands or two bands, making them unsuitable for further Sanger sequencing.

Among the chloroplast primers, five pairs were efficient for the studied crops, namely the 23S,4.5S&5S, 16S, rpl23/rpl2.l, and rpoC1 intron regions, as well as the trnE-UUC/trnT-GUU pair. The greatest number of SNPs and InDels were obtained by rpl23/rpl2.l (fragments of 400—452 bp) and trnE-UUC/trnT-GUU (fragments of 493–686 bp). The lowest numbers of SNPs and InDels were observed in the 16S and rpl2 intron regions. The greatest number of unique SNPs, which is important for the barcoding of cultivars, was observed using rpl23/rpl2.l (Figure 1).

Among the species, actinidia and citrus showed higher levels of polymorphism, which can be explained by the four different species included in the dataset. The lowest level of polymorphisms was observed in the feijoa accessions.

To visualize intraspecific distances revealed by each primer pair, we performed neighbor-joining analysis and reconstructed phylogenetic trees (Figure 2). The results show that most of the chloroplast regions clearly displayed interspecific distances among the genera, except for 16S (Appendix A). Despite the low bootstrap support, almost all primers generated intraspecific polymorphisms for one or several species. Particularly, the trnE-UUC/trnT-GUU pair clearly distinguished between the tea, feijoa, and citrus genotypes (Figure 2A). Principal coordinate analysis (PCoA) revealed 42 and 38% variability in Components 1 and 2, respectively. A close relationship between tea and actinidia accessions was observed, with their related datapoints located on the negative sides of Component 1 and Component 2. Meanwhile, the feijoa and citrus datapoints are respectively located on the positive sides of Component 1 and Component 2, with high loadings. Other chloroplast loci, amplified by rpl23/rpi2.1, revealed clear intraspecific polymorphisms of mandarin genotypes (Figure 2B). Based on neighbor-joining analysis, all crops were separately branched at the species level. Additionally, significant distances were observed among the actinidia and citrus accessions. PCoA analysis demonstrated 57 and 22% variability in Components 1 and 2, respectively. Interestingly, the actinidia datapoints located on the negative sides of both components demonstrate a greater genetic distance from the other crops (Figure 2A).

To clearly represent intraspecific distances in each genus, we further reconstructed separate trees with efficient chloroplast primers (Figure 3). For the actinidia accessions, four chloroplast primer pairs were selected (trnE/trnT (Figure 3A), rpoC1 intron (Figure 3E), 23S,4.5S&5S (Figure 3H), and rpl23/rpi2.1 (Figure 3J)). Among them, the first one clearly distinguished the four species of actinidia included in the study (*A. chinensis* var. *deliciosa*, *A. erantha*, *A. kolomikta*, and *A. arguta*). In addition, cultivar-specific polymorphisms were observed within *A. chinensis* var. *deliciosa* and *A. arguta.* Other three primer pairs displayed mixed branches of these two species.

For the tea crops, three efficient primer pairs with the greatest number of intraspecific polymorphisms were selected (trnE/trnT, rpoC1 intron, and 23S,4.5S&5S). Most of the tea accessions were clearly distinguished from the others based on the polymorphisms in these loci (Figure 3B,F,I). However no separate groups of samples were observed within the *C. sinensis* dataset.

Regarding mandarins, only two regions with significant variability were selected, amplified by trnE/trnT and rpl23/rpi2.1. Based on the neighbor-joining analysis of trnE/trnT loci, the mandarin group showed no intraspecific diversity among satsuma cultivars (*C. unshiu*); however, other mandarins (*C. reshni*, *C. clementina,* and *C. leiocarpa*) were placed on separate branches distinct from satsumas (Figure 3C). In contrast, for the satsuma group, rpl23/rpi2.1 showed efficient cultivar-level separation (Figure 3G).

For feijoa accessions, significant intraspecific distances were also observed by trnE/trnT: several local genotypes were placed on the separate branches (Figure 3D).

## 3. Discussion

The development of an efficient DNA barcoding approach is important for the protection of the copyrights of breeders to protect farmers from buying counterfeit products and enabling them to obtain reliable and complete information about a particular variety. This study was aimed to analyze intraspecific polymorphisms of several loci in chloroplast DNA and the ITS region. The results show significant levels of intraspecific polymorphisms in cpDNA loci, particularly in several introns. Despite the fact that the nuclear ITS region is known for a higher level of polymorphisms compared to cpDNA loci, it is not efficient for highly heterozygous crops, such as tea and citrus. The amplification of the ITS region resulted in two band sizes, making further purification and sequencing difficult. So, for these crops, plastid loci can be a simpler approach for the barcoding of cultivars because the isolation of amplified fragments from the gel is easier and the quality of reads is higher compared to the nuclear ITS region.

In 2009, a large group of plant DNA barcoding specialists proposed using a universal barcode consisting of a combination of *rbcL* and *matK*—two genes encoded in the chloroplast genome. To increase the resolving power, it was proposed to add a third locus—nuclear ITS2 [16,17]. Coding regions such as *rbcL* and *matK* are often preferred for their conservative nature and ease of amplification, while non-coding spacers, such as trnH-psbA and ITS2, are valued for their higher polymorphism. Nevertheless, this barcode was insufficient for intraspecific diversity analysis and has been met with great skepticism since its proposal [18].

To date, several approaches to plant DNA barcoding have been developed using the target region of nuclear DNA (ITS region) and chloroplast DNA (trnH-psbA, rpoB, and rpoC1 genes, etc.) [12,13,14,15]. A high discrimination power of ITS sequences was demonstrated in pineapple [19], cassava [20], banana [21], apricot [22], and mango [23]. However, studies on pomegranate [24] and fig [25] have found the ITS region to be ineffective for distinguishing across cultivars. Limitations of ITS barcoding, such as copy number variation, PCR bias, hybridization, and polyploidy of genomes, make the application of these loci resource-intensive to avoid errors and “noises”. These challenges make the unification of protocols difficult in large-scale experiments. Therefore, compared to the ITS region, cpDNA loci can serve as more efficient barcodes due to the simplicity of their amplification, sequencing, and data analysis, and the greater transferability of primers among different genera.

In 2011, the polymorphism of several plastid loci (rpl23&rpl2.1, 16S, 23S,4.5S&5S, petB&D, and rpl2, rpoC1, and trnK introns) was evaluated in 96 different plant species, It was found that the trnK and rpoC1 introns were the most variable in closely related species, showing high potential as additional barcodes [14]. However, some of the above-mentioned loci provide low resolution in the analysis of intraspecific polymorphisms of several crops [26,27].

Intergenic regions of cpDNA are known for their greater polymorphism compared to genes. According to Dong et al. (2012) [28], analysis of chloroplast genomes in 12 angiosperm genera revealed the top 5% most variable loci. Among the 23 most variable loci (in order from highest to lowest variability) were the intergenic regions ycf1-a, trnK, rpl32-trnL, and trnH-psbA, followed by trnSUGA-trnGUCC, petA-psbJ, rps16-trnQ, ndhC-trnV, ycf1-b, ndhF, rpoB-trnC, psbE-petL, and rbcL-accD. Three loci, trnSUGA-trnGUCC, trnT-psbD, and trnW-psaJ, showed very high nucleotide diversity per site across three genera [28]. Another study reported that the trnH-psbA barcode region showed greater resolving power compared to rbcL and matK in several plant species [29]. However, some intergenic regions are not always efficient enough for cultivar identification. Particularly, the trnH-psbA and trnL-trnF intergenic spacers were able to distinguish and identify only 4 out of 25 *Prunus* accessions [30]. Other research showed that trnH–psbA is not suitable for some taxa due to intraspecific inversions and rps19 insertions, which inflate intraspecific variation [31]. Particularly, trnH-psbA was efficient for distinguishing *Physalis* species, but intraspecific polymorphism was low [32]. In addition, most of the intergenic regions were not efficient for *Triticum* species barcoding. However, a combination of the intergenic regions trnfM-trnT with either trnD-psbM, petN-trnC, matK-rps16, or rbcL-psaI demonstrated a very high discrimination capacity [33]. These results confirm the necessity of crop-specific selection of loci for cultivar barcoding.

## 4. Materials and Methods

### 4.1. Plant Materials and DNA Extraction

For this study, we used plant materials of the core germplasm collection of the Subtropical Scientific Centre (Sochi, Russia, 43.59705546553277, 39.763533606594436) (Table 1). A set of valuable genotypes (mutant forms, local cultivars, and breeding lines) of the core germplasm collections were included in the study. Mature healthy leaves were collected from adult trees and stored at +4 °C. Full DNA was extracted through powdering in liquid nitrogen with the following CTAB protocol [34]. DNA quality was assessed using agarose gel electrophoresis and spectrophotometrically with BioDrop µLite (Biodrop, Cambridge, UK). All samples were diluted to 20 ng µL^−1^ and stored at −20 °C.

### 4.2. DNA Primers and PCR Conditions

Eleven universal primer pairs were selected for the assessment of intraspecific genetic polymorphisms in the abovementioned subtropical crops (Table 2). A 50 μL PCR mixture consisted of 25 μL of 2 × HS-TaqPCR reaction buffer (Biolabmix, Novosibirsk, Russia) containing Hot Start Taq-Polymerase, 0.5 μL of each primer (10µM), 2 μL of DNA (20 ng µL^−1^), and PCR-grade water. Amplification was carried out using the LightCycler^®^ 96 thermal cycler (Roche, Basel, Switzerland) with the following amplification program: one cycle of initial incubation at 95 °C for 4 min; thirty cycles of amplification, including melting at 94 °C for 30s, annealing at 58 °C for 30s, and elongation at 72 °C for 30s; one cycle of final elongation at 72 °C for 7min. The quality of the amplification was assessed by electrophoresis in 2% agarose gel in 1 × TAE buffer for two hours.

### 4.3. Sanger Sequencing and Data Analysis

Enzymatic purification of the amplicons was performed using exonuclease I (ExoI) and alkaline phosphatase (Sap). First, 1u of each enzyme was added to each sample and incubated for 15 min at 37 °C. Then, the enzymes were inactivated for 15 min at 85 °C. Sanger sequencing with the forward primers was performed on an ABI3500xl genetic analyzer (ThermoFisher Scientific, Waltham, MA, USA) using the BrilliantDye v3.1 kit (NIMAGEN, Nijmegen, The Netherlands) following the manufacturer’s instructions. The results were analyzed first using MAFFT add (https://mafft.cbrc.jp/alignment/server/add.html, assessed on 18 July 2025), with the option “Align full-length sequences to an MSA”. The obtained high-quality sequences were aligned in MEGA-X using the Clustal algorithm, followed by reconstruction of the phylogenetic trees using the neighbor-joining algorithm [35,36,37,38].

## 5. Conclusions

The greatest levels of polymorphisms were detected by trnE-UUC/trnT-GUU for all species. In addition, the 23S,4.5S&5S, and rpoC1 intron regions were efficient for tea and actinidia species, while rpl23/rpi2.1 was efficient for citrus and actinidia. Cultivar-level identification was observed using these loci. ITS was not efficient for tea, but displayed reliable results for actinidia. These results provide new perspectives for using universal chloroplast primers for DNA barcoding of valuable cultivars of related species.

## Figures and Tables

**Figure 1 ijms-26-06921-f001:**
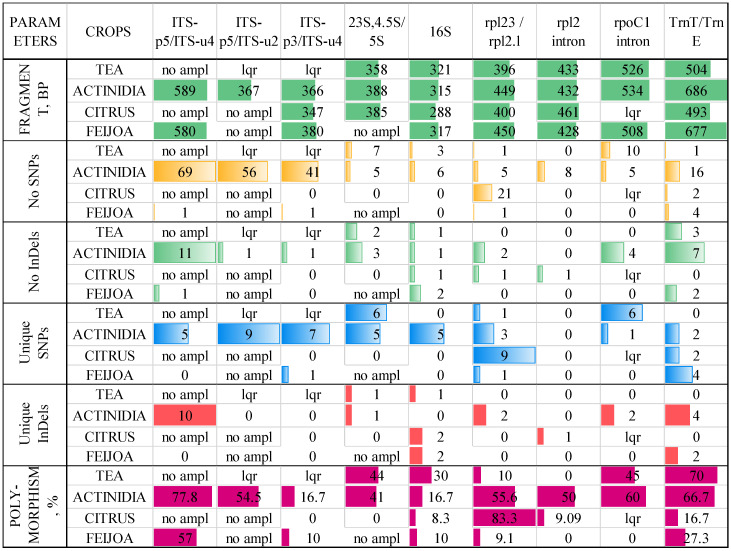
Results of the primer test. Colors in the figure represent conditional formatting to evaluate the difference of the values.

**Figure 2 ijms-26-06921-f002:**
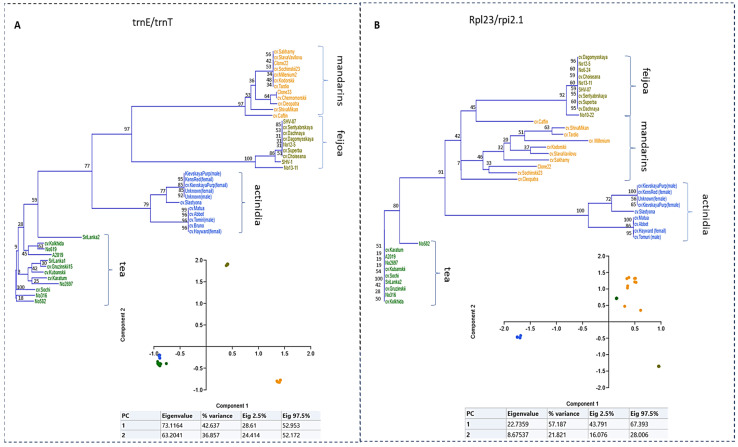
Neighbor-joining and PCoA plots based on SNPs and InDels in the fragments of chloroplast DNA. (**A**) data obtained by amplification with trnE/trnT; (**B**) data obtained by amplification with rpl23/rpi2.1. The colors indicate the related crops: orange—mandarins, blue—actinidia, olive—feijoa, green—tea accessions. The numbers on the branches indicate bootstrap support, with bootstrap values of 1000. The raw sequencing data and alignments can be found in the Appendix A, attached to the article.

**Figure 3 ijms-26-06921-f003:**
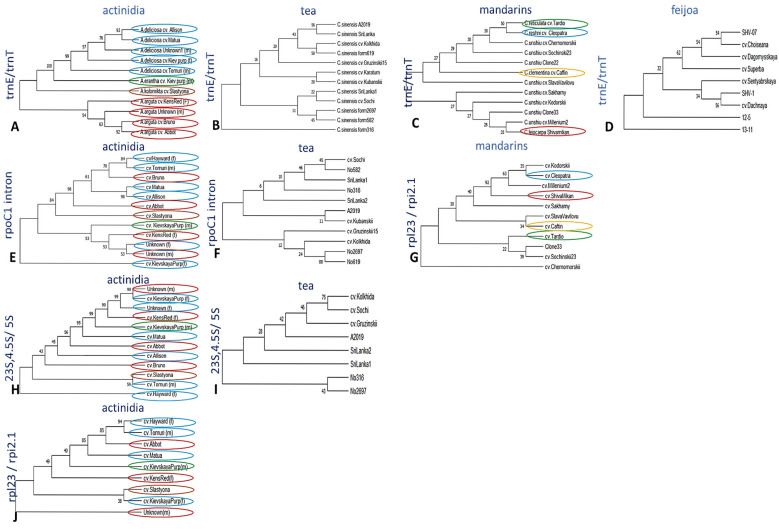
Neighbor-joining analysis of intraspecific diversity based on chloroplast DNA loci: (**A**–**D**) trnE-UUC/trnT-GUU polymorphisms in tea, actinidia, mandarins, and feijoa, respectively; (**E**,**F**) rpoC intron polymorphisms in tea and actinidia; (**G**,**J**) rpl23/rpi2.1 polymorphisms in mandarins and actinidia; (**H**,**I**) 23S,4.5S&5S polymorphisms in tea and actinidia. Different species within actinidia and mandarins are highlighted by colored cycles. The numbers on the branches indicate bootstrap support, with bootstrap values of 1000. The raw sequencing data and alignments can be found in the Appendix A attached to the article.

**Table 1 ijms-26-06921-t001:** Accessions included in the study. The background color indicate four different crops included in the study.

#	Species	Genotype	Origin
1	*Camellia sinensis* (L.) Kuntze	#619	Tetraploid, local form, ɣ-irradiation
2	*Camellia sinensis* (L.) Kuntze	#582	Triploid, local form, ɣ-irradiation
3	*Camellia sinensis* (L.) Kuntze	#2697	Aneuploid, local form, ɣ-irradiation
4	*Camellia sinensis* (L.) Kuntze	cv. Sochi	Diploid, local cultivar, clonal selection
5	*Camellia sinensis* (L.) Kuntze	cv. Kubanskii	Diploid, local cultivar, clonal selection
6	*Camellia sinensis* (L.) Kuntze	cv. Kolkhida	Diploid, local cultivar, clonal selection
7	*Camellia sinensis* (L.) Kuntze	#316	Aneuploid, local form, ɣ-irradiation
8	*Camellia sinensis* (L.) Kuntze	cv. Gruzinskii15	Diploid, clonal selection
9	*Camellia sinensis* (L.) Kuntze	SriLanka1	Diploid, clonal selection
10	*Camellia sinensis* (L.) Kuntze	SriLanka2	Diploid, clonal selection
11	*Camellia sinensis* (L.) Kuntze	cv. Karatum	Triploid, local cultivar
12	*Camellia sinensis* (L.) Kuntze	A2019	Diploid, local breeding line
13	*Actinidia chinensis* var. *deliciosa* (A.Chev.) A. Chev.	Tomuri (male)	New Zealand
14	*Actinidia chinensis* var. *deliciosa* (A.Chev.) A. Chev.	Hayward (female)	New Zealand
15	*Actinidia arguta* (Siebold & Zucc.) Planch. ex Miq.	Kievskaya Purpurnaya (female)	Ukraine
16	*Actinidia arguta* (Siebold & Zucc.) Planch. ex Miq.	Unknown (male)	Russia
17	*Actinidia kolomikta* (Maxim. & Rupr.) Maxim.	Slastyona (female)	Russia
18	*Actinidia arguta* (Siebold & Zucc.) Planch. ex Miq.	Unknown (female)	Russia
19	*Actinidia arguta* (Siebold & Zucc.) Planch. ex Miq.	Ken’s Red (female)	New Zealand
20	*Actinidia arguta* (Siebold & Zucc.) Planch. ex Miq.	Kievskaya Purpurnaya (male)	Ukraine
21	*Actinidia chinensis* var. *deliciosa* (A.Chev.) A. Chev	Bruno (female)	New Zealand
22	*Actinidia chinensis* var. *deliciosa* (A.Chev.) A. Chev	Abbott	New Zealand
23	*Actinidia chinensis* var. *deliciosa* (A.Chev.) A. Chev	Allison	New Zealand
24	*Actinidia chinensis* var. *deliciosa* (A.Chev.) A. Chev	Matua	New Zealand
25	*Citrus reshni* hort. ex. Tanaka	Cleopatra	India
26	*Citrus reticulata* Blanco.	Tardio	Italy
27	*Citrus unshiu* Marcow. × *C.* × *leiocarpa*	Chernomorskii	Local cultivar
28	*Citrus unshiu* Marcow.	Sochinskii23	Local cultivar
29	*Citrus unshiu* Marcow.	Clone22	Local cultivar
30	*Citrus reticulata* var. *austera* Swingle	Caftin	USA
31	*Citrus unshiu* Marcow.	Slava Vavilovu	Local cultivar
32	*Citrus unshiu* Marcow.	Sakharny	Local cultivar
33	*Citrus unshiu* Marcow.	Kodorskii	Local cultivar
34	*Citrus unshiu* Marcow.	Clone33	Local cultivar
35	*Citrus leiocarpa* hort. ex Tanaka	Shiva Mikan	India
36	*Citrus unshiu* Marcow.	Millenium2	Local cultivar
37	*Feijoa sellowiana* (O. Berg) Burret	#6-24	Local breeding line
38	*Feijoa sellowiana* (O. Berg) Burret	#13-11	Local breeding line
39	*Feijoa sellowiana* (O. Berg) Burret	#10-22	Local breeding line
40	*Feijoa sellowiana* (O. Berg) Burret	Choiseana	USA
41	*Feijoa sellowiana* (O. Berg) Burret	Dachnaya	Local cultivar
42	*Feijoa sellowiana* (O. Berg) Burret	Superba	USA
43	*Feijoa sellowiana* (O. Berg) Burret	Sentyabrskaya	Local cultivar
44	*Feijoa sellowiana* (O. Berg) Burret	SHV-07	Local breeding line
45	*Feijoa sellowiana* (O. Berg) Burret	#12-5	Local breeding line
46	*Feijoa sellowiana* (O. Berg) Burret	Dagomysskaya	Local cultivar
47	*Feijoa sellowiana* (O. Berg) Burret	SHV-1	Local breeding line

**Table 2 ijms-26-06921-t002:** Primers selected for the study.

Primers Code	Primer Sequences 5′–3′	Product Length, bp	Amplified Region	Reference (doi)
ITS-P5/ITS-U4	F-ccttatcayttagaggaaggag; R-rgtttcttttcctccgctta	700–800	nuclear genome, ITS-region	[14]
ITS-P5/ITS-U2	F-ccttatcayttagaggaaggag; R-gcgttcaaagaytcgatgrttc	No data	nuclear genome, ITS-region	[14]
ITS-P3/ITS-U4	F-ygactctcggcaacggata; R-rgtttcttttcctccgctta	410–480	nuclear genome, ITS-region	[14]
23S,4.5S/5S	F-tctcctccgacttccctag; R-accatgaacgaggaaaggc	400–430	chloroplast genome	[14]
16S	F-attgcgtcgttgtgcctgg; R-gatacgttgttaggtgctcc	350–370	chloroplast genome	[14]
PetB/PetD	F-tagggggaattacacttac; R-cattaacatgaatacggcag	490–500	chloroplast genome	[14]
rpl23/rpl2.L	F-gaagaagcttgtacagtttgg; R-tttacttacggcgacgaag	490–500	chloroplast genome	[14]
rpl2 intron	F-attgagttcagtagttcctc; R-ccaaactgtacaagcttcttc	430–520	chloroplast genome	[14]
RPOC1 intron	F-gagtaacatgaagctcag; R-gtttcctttcatccggct	540–670	chloroplast genome	[14]
TRNK intron	F-gtctacatcatcggtagag; R-caacccaatcgctcttttg	430–500	chloroplast genome	[14]
TRNE-UUC/TRNT-GUU	F-tcctgaaccactagacgatg; R-atggcgttactctaccactg	834	chloroplast genome	[17]

## Data Availability

All data will be made available on request. Contact email: q11111w2006@yandex.ru (L.S.S.).

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
