# Peer review of "The Potential of Universal Primers for Barcoding of Subtropical Crops: Actinidia, Feijoa, Citrus, and Tea"

_ijms, 2025, doi:10.3390/ijms26146921_

Round 1

Reviewer 1 Report

Comments and Suggestions for Authors

In the manuscript “Intraspecific structural polymorphisms obtained by universal primers provide great potential for barcoding of subtropical crops (actinidia, feijoa, citrus and tea)” by Samarina et al., the author The author used Sanger sequencing to evaluate 11 universal primers ITS-p5/ITS-u4, ITS-p5/ITS-u2, ITS-p3/ITS-u4, 23S, 4.5S/5S, 16S, petB/petD, rpl23/rpl2.l, rpl2, the potential of DNA barcoding in 47 germplasms (superior varieties, variants, and breeding lines) of subtropical crops such as kiwifruit, flying flower, citrus, and tea, including introns rpoC1, trnK introns, and trnC UuC/trnT GUU. The results showed that among these primers, rpl23/rpl2. l and trnE UUC/trnTGUU exhibited greater intraspecific polymorphism for all crops, while the rpl2 intron and 16S showed the lowest levels of polymorphism in the generated fragments. These results will be useful for the further development of DNA-barcodes of the related species.

The entire paper has clear logic, clear analysis of results, and beautiful image production. Moreover, I have some other concerns, listed as following:

  1. In terms of material methods, it is best for the author to briefly introduce the geographical location information of the plant sample collection site, the growth environment conditions of the plant material, and when the sample was collected.
  2. The discussion section needs to be optimized by conducting in-depth discussions based on the results of this study, analyzing and comparing the differences and advantages between the results of this study and previous research, in order to better reflect the significance of this study.
  3. The content in the conclusion should not be repeated with the abstract, and the author needs to further optimize this part.
  4. The author needs to pay attention to the way specialized nouns are expressed, for example, the Latin name of a plant needs to be italicized. In addition, the author should follow the formatting requirements of the journal for formatting, especially for references.

Author Response

Comment 1: [In terms of material methods, it is best for the author to briefly introduce the geographical location information of the plant sample collection site, the growth environment conditions of the plant material, and when the sample was collected.]

Response 1: [in the Table 1 we have added the column with the origin of the genotypes. Also we added information on the collection on MM.]

Comment 2: [The discussion section needs to be optimized by conducting in-depth discussions based on the results of this study, analyzing and comparing the differences and advantages between the results of this study and previous research, in order to better reflect the significance of this study.]

Response 2: [We have completed the section]

Comment 3: [The content in the conclusion should not be repeated with the abstract, and the author needs to further optimize this part.]

             Response 3: [We have shortened the conclusion]

Comment 4: [The author needs to pay attention to the way specialized nouns are expressed, for example, the Latin name of a plant needs to be italicized. In addition, the author should follow the formatting requirements of the journal for formatting, especially for references.]

             Response 4: [Latin names are revised in the table and in the text. The references are revised.]

Reviewer 2 Report

Comments and Suggestions for Authors

The manuscript contains useful data, but several aspects require improvement. Please find my comments below:

  1. Title: The current title is too long and slightly misleading. As I understand, only genetic polymorphism was analysed, not structural (you did not assay the 3D structure of the genes). I recommend shortening the title, for example: "The potential of universal primer barcoding for subtropical crops: Actinidia, Feijoa, Citrus, and Tea."
  2. Data availability: The sequence data must be accessible to readers. Please upload the sequences to GenBank and provide the corresponding accession numbers in the manuscript. Alternatively, you may include the FASTA-formatted alignments as supplementary material.
  3. Formatting and consistency (throughout the manuscript):

    - Latin species names must be italicised everywhere.

    - Use scientific genus names instead of common names.

    - Correct “SNP s / InDels” to “SNPs / InDels.

    - Numbers below 10 should be written as words (e.g., three instead of 3).

    - Lines 66–67: Replace “structural” with “genetic.”

    - Clarify whether sequencing was performed from both primers or only one.
  4. Table 1: Consider adding information about the origin or background of the genotypes used.

  5. Line 107: Specify whether you used PCA or PCoA, and provide the full form of the abbreviation at first mention.

  6. Line 110: This line must be removed.

  7. Figure 1: Ensure that the sample labelling in Table 1 and Figure 1 is consistent.  Add bootstrap support values to the phylogenetic tree. Improve label readability and increase resolution. Each subfigure must have a unique identifier (e.g., A, B, C) and be cited appropriately in the text.

  8. Figure 2: Clearly explain each part of the figure and add proper labelling. Resolution is too low for both the PCoA plot and trees—please improve. The alignment section is currently unreadable and may be omitted. For the PCoA, indicate the explained variance for each axis. Add bootstrap support to the tree. Consider using IQ-TREE and iTOL web tools for constructing and visualising phylogenetic trees (it is only a recommendation).

  9. Line 167: The sentence can be removed.

  10. Line 170: The term “reads” is more appropriate for NGS data; please use “sequences” here.

  11. Conclusion: Remove the first and second sentences to make the conclusion more focused.

Author Response

Comment 1: [Title: The current title is too long and slightly misleading. As I understand, only genetic polymorphism was analysed, not structural (you did not assay the 3D structure of the genes). I recommend shortening the title, for example: "The potential of universal primer barcoding for subtropical crops: Actinidia, Feijoa, Citrus, and Tea."]

Response 1: [revised accordingly]

Comment 2: [Data availability: The sequence data must be accessible to readers. Please upload the sequences to GenBank and provide the corresponding accession numbers in the manuscript. Alternatively, you may include the FASTA-formatted alignments as supplementary material.]

             Response2: [We have attached all sequencing data with identifiers]

Comment 3:[Latin species names must be italicised everywhere.]

Response 3: [revised accordingly]

Comment 4: [Use scientific genus names instead of common names.]

Response 4: [If we use the genus names it will lead to misunderstanding of “intraspecific diversity” term. If we use species names, it will also be difficult because several species of Actinidia and Several species of Citrus are included in the study. That is why we use the name of crop (mandarin, tea, actinidia and feijoa)]

Comment 5: [Correct “SNP s / InDels” to “SNPs / InDels.”]

Response 5: [revised]

Comment 6: [Numbers below 10 should be written as words (e.g., three instead of 3).]

Response 6: [sorry, I have not found these in the text]

Comment 7: [Lines 66–67: Replace “structural” with “genetic.”]

Response 7: [replaced everywhere]

Comment 8: [Clarify whether sequencing was performed from both primers or only one.]

Response 8: [ for only forward primers, added line 103]

Comment 9: [Table 1: Consider adding information about the origin or background of the genotypes used.]

              Response 9: [The column is added]

Comment 10: [Line 107: Specify whether you used PCA or PCoA, and provide the full form of the abbreviation at first mention.]

Response 10: [Revised accordingly]

Comment 11: [Line 110: This line must be removed.]

Response 11: [revised ]

Comment 12: [Figure 1: Ensure that the sample labelling in Table 1 and Figure 1 is consistent.  Add bootstrap support values to the phylogenetic tree. Improve label readability and increase resolution. Each subfigure must have a unique identifier (e.g., A, B, C) and be cited appropriately in the text.]

Response 12: [samples labels are revised on Fig1. Bootstrap values are added. The figure labelling is revised.]

Comment 13: [Figure 2: Clearly explain each part of the figure and add proper labelling. Resolution is too low for both the PCoA plot and trees—please improve. The alignment section is currently unreadable and may be omitted. For the PCoA, indicate the explained variance for each axis. Add bootstrap support to the tree. Consider using IQ-TREE and iTOL web tools for constructing and visualising phylogenetic trees (it is only a recommendation).]

Response 13: [We removed alignment files and chromatograms. We have added new plots based on the other informative loci. The bootstrap consensus trees are constructed. ]

Comment 14: [Line 167: The sentence can be removed.]

Response 14: [removed]

Comment 15: [Line 170: The term “reads” is more appropriate for NGS data; please use “sequences” here.]

Response 15: [revised]

Comment 16: [Conclusion: Remove the first and second sentences to make the conclusion more focused.]

Response 16: [removed]

Round 2

Reviewer 2 Report

Comments and Suggestions for Authors

There is "5" left in the abstract and no number of used bootstrap replicates is indicated. Some bootstrap values are not readable in the Figure 1 and are absent for some nodes. Check the spaces in the text - some are missing, especially pay attention to the references to the Figures in the main text.

Otherwise, authors significanyly improved the manuscript.

Author Response

Comment 1: [There is "5" left in the abstract and no number of used bootstrap replicates is indicated.]

Response 1: [Dear reviewer, we are grateful for your time and work with our MS. The abstract is revised, the bootstrap values are added in the legends of the figures]

Comment2: [Some bootstrap values are not readable in the Figure 1 and are absent for some nodes.]

Response 2: [revised, the missing values are added in the fig.1, the letters size is decreased, please check]

Comment 3: [Check the spaces in the text - some are missing, especially pay attention to the references to the Figures in the main text.]

Response 3: [ checked]